# Combined Systematic Review and Transcriptomic Analyses of Mammalian Aquaporin Classes 1 to 10 as Biomarkers and Prognostic Indicators in Diverse Cancers

**DOI:** 10.3390/cancers12071911

**Published:** 2020-07-15

**Authors:** Pak Hin Chow, Joanne Bowen, Andrea J Yool

**Affiliations:** Adelaide Medical School, University of Adelaide, Adelaide SA 5005, Australia; pakhin.chow@adelaide.edu.au (P.H.C.); joanne.bowen@adelaide.edu.au (J.B.)

**Keywords:** water channel, AQP, metastasis, transcriptomics, forest plot, prognosis, patient survival

## Abstract

Aquaporin (AQP) channels enable regulated transport of water and solutes essential for fluid homeostasis, but they are gaining attention as targets for anticancer therapies. Patterns of AQP expression and survival rates for patients were evaluated by systematic review (PubMed and Embase) and transcriptomic analyses of RNAseq data (Human Protein Atlas database). Meta-analyses confirmed predominantly negative associations between AQP protein and RNA expression levels and patient survival times, most notably for AQP1 in lung, breast and prostate cancers; AQP3 in esophageal, liver and breast cancers; and AQP9 in liver cancer. Patterns of AQP expression were clustered for groups of cancers and associated with risk of death. A quantitative transcriptomic analysis of AQP1-10 in human cancer biopsies similarly showed that increased transcript levels of AQPs 1, 3, 5 and 9 were most frequently associated with poor survival. Unexpectedly, increased AQP7 and AQP8 levels were associated with better survival times in glioma, ovarian and endometrial cancers, and increased AQP11 with better survival in colorectal and breast cancers. Although molecular mechanisms of aquaporins in pathology or protection remain to be fully defined, results here support the hypothesis that overexpression of selected classes of AQPs differentially augments cancer progression. Beyond fluid homeostasis, potential roles for AQPs in cancers (suggested from an expanding appreciation of their functions in normal tissues) include cell motility, membrane process extension, transport of signaling molecules, control of proliferation and apoptosis, increased mechanical compliance, and gas exchange. AQP expression also has been linked to differences in sensitivity to chemotherapy treatments, suggesting possible roles as biomarkers for personalized treatments. Development of AQP pharmacological modulators, administered in cancer-specific combinations, might inspire new interventions for controlling malignant carcinomas.

## 1. Introduction

Membrane channels and transporters are essential for the balanced control of ion and fluid homeostasis and electrical signaling [1] and serve key roles in cell proliferation, migration, apoptosis and differentiation, which are increasingly being recognized as relevant for cancer progression [2]. Aquaporins are known as channels that facilitate passive water transport in response to osmotic gradients created by active transport and net displacement of solutes across cell membranes or tissue barriers [3]. Several classes of mammalian aquaporin (AQP) channels have been linked to cancer progression via effects on angiogenesis, proliferation and metastasis [4]. AQPs have been proposed as part of the volume regulatory engine driving process extension during motility [5,6,7,8]. Inhibitors of AQPs are of interest as potential tools for impairing the protrusion and displacement steps in metastatic cell movement [8]. Pharmacological modulators of aquaporin channels derived from loop diuretics, metal-containing organic compounds, plant natural products and other small molecules are opening opportunities to explore the therapeutic potential of AQPs as novel targets [8].

AQP overexpression has been reported in at least 12 different tumor cell types [9]. Interestingly, different classes of AQPs are upregulated for different cancer types, in which tumor-promoting effects are not reproduced by substitution of other AQP classes [10]. Positive associations between histological tumor grades and levels of AQP expression often involve AQP subtypes in the pathology that are not expressed in normal tissues at the site of origin [11]. This reliance on an AQP subtype indicates that the tumorigenic roles are unlikely to stem simply from increasing water channel activity and osmotic water flux, but must be exploiting features such as substrate permeability, mechanisms of regulation, subcellular localization or other properties that differ between subtypes. AQPs have been proposed to promote cancer metastasis by facilitating tumor cell migration [6,7,12], but a broader range of possible functions is likely, given the expanding understanding of their diverse roles in normal cell physiology.

Thirteen classes (AQP0–AQP12) have been identified in higher mammals [13,14], expressed in kidneys, lung airways, eyes, brain, glands, vascular system and other tissues [15,16]. Classical aquaporins (AQPs 0, 1, 2, 4, 5, 6 and 8) were initially viewed as water-selective channels, but further evidence has established capacity for transport of gases, urea, hydrogen peroxide, ammonia and, in some cases, charged particles [17,18]. Aquaglyceroporins (AQPs 3, 7, 9 and 10) are permeable to glycerol as well as water. AQPs 11 and 12 are more distantly related to other mammalian AQPs, based on amino acid sequence; functions and regulation remain to be fully defined [19]. Diverse contributions of AQP classes could augment pro-cancer conditions by enhancing signaling via nitric oxides or hydrogen peroxide [20,21,22,23], facilitating gas exchange of O_2_ and CO_2_ [24,25], promoting cell cycle progression [26], providing metabolic support [27,28], mediating dual ion and water fluxes for localized control of process extension [12], driving angiogenesis [29], boosting mechanical compliance needed for rapid volume changes [6,30] and other processes. Dynamic translocation between membrane and intracellular vesicle pools is controlled by signaling; for example, AQP2 localization depends on cyclic AMP-dependent kinase activity [31], and AQPs are associated via protein–protein interactions into signaling complexes [32,33,34]. Apoptotic responses associated with AQPs 4, 8 and 9 could involve water and monovalent ion loss, causing the hallmark cell shrinkage which precedes programmed cell death [35,36,37], suggesting AQPs in some cases might also have anticancer effects depending on the balance of outcomes.

Strong evidence for AQPs in cancer cell migration and metastasis comes from in vitro pharmacological studies, as well as in vivo mouse models [7,8,38]. In the process of metastasis, cancer cells escape normal control mechanisms, invade surrounding tissues and spread to other parts of the body, accounting for the second greatest cause of mortality globally, as ranked by the World Health Organization, based on 9.6 million deaths in 2018 [39]. As the average human lifespan lengthens, the incidence of cancer in the aging population is increasing, with numbers expected to rise by 70% in the next two decades [40]. Treatment options, including surgery, chemotherapy and radiotherapy, are aimed primarily at inhibiting cancer proliferation [41], now recognized as being mediated in part by activation of the immune system, and driving new interest in combined immunotherapies [42]. However, the recurrence of cancers at new sites indicates that additional treatments targeting cancer metastasis are greatly needed [43].

The hypothesis tested in this work was that levels of expression of different classes of AQPs (based on data for transcript levels, protein levels or both) show distinctive patterns that are associated with the risk of death in people with cancers. The combined analyses here are the first, to our knowledge, to systematically review all classes of aquaporins in diverse cancer types and to correlate findings with transcriptomic data from human cancer biopsies.

## 2. Results

A total of 1546 papers were identified from the first level of screening of PubMed (https://pubmed.ncbi.nlm.nih.gov/) and Embase (https://www.embase.com/) databases by using the search keywords (Table 1). After excluding duplicate search results, 361 records were selected for second-level screening of titles and abstracts. After careful assessment, 285 records were retrieved for full review. Papers lacking sufficient focus on AQPs were excluded (with reasons logged), leaving a final set of 224 papers included in this review (Figure 1). Within this final set, the largest proportion of studies addressed AQP1, followed by AQP5, AQP4, AQP3 and AQP9, for a diverse array of cancer types, including brain, lung, breast and colorectal cancers.

Forest plots (Figure 2) summarizing the survival probabilities of people with cancers reported in the published literature were correlated with levels of expression of different classes of AQPs (based on RNA, protein levels or both), as determined from compiled data from all papers in the final set, which included survival analyses (n = 30). Results indicated strongly negative correlations. AQP1 appeared to be associated with higher risks of death in lung adenocarcinoma patients with a four-fold increase in hazard ratio (HR 4.0) and in pleural mesothelioma (HR 2.7), as well as breast, prostate and some colon cancers (HRs 2.6 to 3.4). Dramatic increases in patient risk were observed for AQP3 in esophageal cancer (HR18.4), and AQP9 in liver cancer (HR 10.8). High hazard ratios for gastric cancer patients correlated with increased AQPs 2, 8 and 10 expression, contrasting with a reduced hazard ratio seen when AQP3 or AQP9 levels were increased. For breast cancer patients, higher hazard ratios were observed with increased AQPs 1 and 3 expression; possible associations with other AQP classes remain to be evaluated. These data support the idea that AQPs are upregulated in cancers and that the specific classes of AQPs involved and patterns of co-regulation depend on the cancer subtype [10]. It is important to note that, in some cases, such as hepatocellular carcinoma, increased expression of AQP1 is associated with the vasculature, and rarely the cancer cells themselves [44].

The finite number of publications in this diverse field preclude an exhaustive comparison of all classes of AQPs across all cancer types. In parallel with the systematic review data, an independent analysis of AQP RNA expression levels by cancer type was carried out for glioma, colorectal, lung, breast, ovarian and endometrial cancers, using RNAseq transcriptomic data compiled from the Human Protein Atlas database, to serve as a second arm of this study. Transcriptomic outcomes were compared with the results of the systematic review, as summarized in the sections below, to identify corroborating lines of evidence, inconsistent findings and interesting gaps in knowledge that could merit future research.

The three limitations of this study are as follows: (i) Data from diverse experimental methods and models were merged from the published papers that were included in the systematic analysis; (ii) the number of published studies to date regarding aquaporin channels in cancers is limited; and (iii) RNAseq transcriptomic data are based on human biopsy samples taken at random locations in excised human tumor masses, and thus do not necessarily reflect a full array of gene transcripts that might exist in heterologous assemblies of cancers cells located in different regions of tumor masses. In sum, these limitations means that AQPs which are identified as important for cancer progression in one arm of analysis here might not necessary be mirrored perfectly by data compiled in the complementary analysis, and that absence of evidence does not rule out potential roles for these classes of AQP channels in cancers that are yet to be investigated.

### 2.1. AQPs in Gliomas

The human brain consists of 100 billion neurons and one trillion glial cells, on average [61]. Localized in astrocyte end feet throughout the brain and spinal cord, AQP4 enables central nervous system fluid homeostasis and promotes maintenance of the blood–brain barrier [62]. AQP1 normally is expressed in the choroid plexus epithelium, where it contributes to cerebral spinal fluid secretion by mediating water flux from blood to brain [63,64], but it is otherwise not abundant in healthy brain tissues. AQP9, a channel permeable to a variety of organic substrates, including lactate, glycerol and other solutes, is expressed at low levels in glia and neurons, where it is speculated to play a role in energy metabolism, though details remain to be clarified [65].

Upregulation of AQP1 and AQP4 protein and RNA has been the major focus of papers published in the glioma field (Table 2), with additional work identifying possible involvement of AQP9. AQP1 overexpression has been observed in diverse types of gliomas, such as glioblastoma, astrocytomas, oligodendrogliomas, ependymomas and glioastrocytomas; the levels of expression have been reported to correlate with the grade of malignancy and invasiveness of the tumors [66,67,68,69,70,71,72]. Glioma invasiveness has been linked to AQP1 overexpression [73], which is greater in migrating cells than in the tumor core [68]. Dexamethasone, which promotes AQP1 transcription, increases the invasiveness of glioma cells [74,75].

AQP4 upregulation and redistribution in glioblastoma [82,93,95,100,105,111] has been suggested to contribute to tumor-associated edema observed by magnetic resonance imaging [91,112,113], and conversely enhance clearance of excess fluid [103]. Genetic deletion of AQP4 impairs cell migration, actin polymerization and apoptosis [89,90]. Downregulation of AQP4 expression in glioma by pentamidine and temozolomide promoted apoptosis and inhibited cell migration, which could be a potential treatment for glioma [87,88]. The incidence of epileptiform seizures in glioma patients correlated with increased membrane levels of AQP4 protein, though transcript levels were not altered [114], raising the important point that not just translational synthesis but also subcellular localization of proteins is essential for deciphering functional outcomes. AQP4 isoforms are able to form heterotetramers, which can turn assemble into higher-order structures called orthogonal arrays of particles, indicating a link to actin cytoskeleton [115]. AQP9 is not highly expressed in normal brain; however, increased levels observed in human glioma were correlated with pathological grade [108] and are proposed to promote cell invasiveness via an AKT signaling pathway [110].

Separate analyses of transcriptomic data from the Human Protein Atlas pathology database complemented the systematic literature review that was aimed at assessing links between AQP transcript levels and cancer risk outcomes. Transcript levels in human gliomas ranged up to high levels for AQP1 and AQP4 (Figure 3A), whereas other classes of AQPs generally showed slightly lower levels, or no difference, as compared to overall AQP median values. The risk of death for glioma patients after diagnosis (Figure 4A) showed moderate possible associations with transcript levels for AQPs 1, 2, 3, 6, 9 and 11 (HRs ranging from 1.2 to 1.6), but the hazard ratio was almost doubled for patients with AQP5 transcript levels exceeding the median value (HR1.9). AQP4 was not associated with a negative effect (HR 0.9), perhaps not surprising given that AQP4 is ubiquitously expressed at high levels in normal glia. There is a substantial gap in knowledge on possible roles of AQPs 5 and 9, which showed the highest hazard ratios but are comparatively underexplored in published work, suggesting an area that merits future study in glioma research.

### 2.2. AQPs in Colon Cancer

Colorectal (bowel) cancer initiates from small noncancerous polyps inside the colon and mainly affects older adults [39]. In normal colon tissue, AQPs 1, 3, 4, 7 and 8 are the predominant isoforms, which are responsible for water absorption [116]. The expression and functions of AQP1 and AQP5 have been the main focus of papers published in the colorectal cancer field (Table 3). AQP1 is upregulated from early through late stages of colorectal carcinogenesis, and expression levels have been suggested to correlate with tumor invasiveness, prompting classification of AQP1 as a negative prognostic indicator of patient survival [56,117,118]. Molecular knockdown and pharmacological inhibition of AQP1 in colon cancer cells significantly impaired migration, supporting AQP1 as a candidate target for colon cancer therapy [118,119,120,121,122,123]. Effects on motility could arise from AQP1-associated effects on actin organization via RhoA and Rac signaling pathways [123]. Alternatively, the role of AQP1 in colon cancer might depend on a dual water and ion channel function suggested to promote lamellipodial extension and cell migration [5,12,123]. AQP5 similarly has been proposed as a prognostic biomarker for colorectal cancer, with AQP5 levels found to be proportional to numbers of circulating tumors cells [124,125,126] and risk of liver metastases [126]. Cell proliferation induced by increased AQP5 involved Ras-MAPK signaling pathways [126]; conversely, AQP5 knockdown inhibited proliferation and triggered apoptosis [127,128]. AQP3 has been suggested to modulate tumor differentiation in colon cancer patients via the EGFR pathway, but its role remains to be defined [129]. Glycerol-permeable AQP3 in other tissues is involved in nutrient uptake and metabolism [28], and it could contribute similarly in some cancers.

In related work, levels of AQPs 1, 5 and 9 protein expression in biopsied samples have been associated with the effectiveness of chemotherapy applied post-surgery in patients with Stages II and III colorectal carcinoma, suggesting another use for AQPs as biomarkers in personalized medicine [140]. Genetic knockdown of AQP5 increased sensitivity to chemotherapy and downregulated p38 MAPK signaling in colon cancer cells [127,134]. Conversely, colon cancer patients non-responsive to adjuvant chemotherapy were more likely to have low AQP9 expression [137,138].

In agreement with one of the studies from the systematic review summary shown in Figure 2 AQP transcript data extracted from the Human Protein Atlas database indicated elevation of AQP1 RNA in colorectal cancers (Figure 3B), as referenced to overall median AQP. AQP3 transcript levels also appeared high, suggesting a gap in knowledge with regard to a possible role in colon cancer. However, the patterns of expression of AQPs showed no association with survival time for any of the classes of AQPs (Figure 4B), suggesting that although classes of AQPs might enable important aspects of cancer progression, they fall short of serving as robust negative prognostic indicators for survival in colorectal cancer patients, based on data available to date.

### 2.3. AQPs in Lung Cancer

Lung cancers can be divided into two main groups, small-cell lung cancer (SCLC) and non-small-cell lung cancer (NSCLC), which can be further subdivided into adenocarcinoma, squamous cell carcinoma and large-cell carcinoma [141,142,143]. In lung tissues, AQP 1 protein normally is expressed in microvascular endothelia, and AQP3 and AQP4 are in airway epithelia [144]. Overexpression of AQP1, AQP3 and AQP5 has been the major focus of papers published in the lung cancer field (Table 4). AQP1 protein and RNA expression, upregulated in lung adenocarcinoma and bronchoalveolar carcinoma (but not in lung squamous cell carcinoma), was correlated with a high risk of postoperative metastasis and low disease-free survival rates, and therefore suggested to be a prognostic factor for stage categories and histologic differentiation of lung cancers [46,142,143,145,146]. Poor survival in asbestos-related mesothelioma similarly was associated with higher AQP1 levels identified in neoplastic tissues by immunocytochemistry [45,48]. Interestingly, chemotherapy appeared to result in increased AQP1 expression [147], which might, in theory, cause a counterproductive boost in cancer recurrence by enhancing invasiveness. Transfection of AQP1 into lung cancer cells enhanced proliferation in vitro [145]; AQP1 overexpression in capillary endothelia of lung adenocarcinoma and mesothelioma tumors promoted angiogenesis, which would facilitate cancer growth and spread [146]. Levels of AQP3 in lung adenocarcinoma were correlated with tumor differentiation and clinical stage [129]. Increased AQP3 also promoted angiogenesis in lung cancer through HIF-2α–VEGF, and invasion via AKT–MMP pathways [148]. In NSCLC patients, high levels of AQP4 did not correlate with poorer survival [101]; in contrast, AQP5 overexpression was associated with unfavorable outcomes [149,150,151]. Genetic knockdown of AQP5 in cell lines reduced migration [152,153], whereas upregulation of AQP5 was associated with activated epidermal growth factor receptor (EGFR), extracellular receptor kinase (ERK1/2) and p38 mitogen-activated protein kinase (p38 MAPK) signaling pathways, to facilitate proliferation and migration [154]. AQP3 coexpression with AQP5 was linked to poor survival, suggesting that combined detection of markers might strengthen prognostic predictive value [58].

Consistent with systematic review results in Figure 2 showing a high hazard ratio for AQP1 (HR 4), increased transcript levels for AQP1 were observed in lung cancer biopsies (Figure 3C). A parallel increase in levels of AQP3 could point to a gap in knowledge regarding a possible role in lung cancer. However, the hazard ratio calculations for overall survival time of patients with lung cancer showed no convincing associations with expression levels of any classes of AQPs, apart from a possible small increase in hazard (HR 1.3) with AQP9 (Figure 4C), suggesting another area for future inquiry. The potential clinical value of any of the classes of AQPs as reliable prognostic indicators in lung cancer thus remains unclear at present.

### 2.4. AQPs in Breast Cancer

Breast cancer adenocarcinoma begins with mutated cells in the milk ducts or lobules [168]. Upregulation of AQPs 1, 3 and 5 has been the major focus of papers published in the breast cancer field (Table 5). AQP1 upregulation in breast cancer, induced by estrogen [169] and negatively regulated by microRNA-320 [170], was correlated with prognoses of poor survival for breast cancer patients [51,171]. High levels of AQP1 protein were associated with the most aggressive subtypes of basal-like breast carcinomas [49]. Protein levels of AQP1 similarly were correlated with poor outcomes, high rates of recurrence and invasiveness in other cancers, including prostate adenocarcinoma [50], biliary tract carcinoma [172] and gastric cancer [55]. AQP3 is upregulated in the early stages of breast cancer, in response to fibroblast growth factor via FGFR–PI3K or FGFR–ERK signaling pathways, and estrogen [173,174]. High AQP3 expression is correlated with low patient survival rates post-surgery, suggesting value as a prognostic marker [47,57]. Increased levels of AQP3 channels, mediating H_2_O_2_ transport and inducing CXCL12- cell signaling and migration, could promote breast cancer metastasis [174,175]. Following a similar pattern, AQP5 upregulation by estrogen in breast cancer patients from early stages correlated with reduced survival times, suggesting AQP5 also was a prognostic marker [176,177]. Knockdown of AQP5 activated the MAPK signaling pathway, reducing cell invasiveness and proliferation, and enhanced the chemosensitivity of breast cancer cells, suggesting AQP5 is of interest as a biomarker and a pharmacological target [178,179].

Results from the systematic review shown in Figure 2 indicated that increased AQPs 1 and 3 protein and RNA expression levels correlated with increased risk of death with hazard ratios of 2.9 and 3.1, respectively. In agreement, transcriptomic analyses from the Human Protein Atlas database showed distinctly higher levels for AQP1 and AQP3 transcripts (Figure 3D), with AQPs 5, 7, 9 and 11 also showing possible moderately increased levels, as compared with overall median AQP levels in breast cancer biopsies. However, apart from AQP3 with a hazard ratio of 1.5, the patterns of upregulated AQP expression were not correlated with reduced survival rates in breast cancer patients (Figure 4D), based on data available to date. Interestingly, increased levels of AQPs 4 and 11 were linked to longer overall survival times (HR 0.6 in both), suggesting some classes of AQP subtypes might have an anticancer potential yet to be defined.

### 2.5. AQPs in Ovarian Cancer

AQPs 1, 2, 3 and 4 channels are normally expressed in ovary [185,186]. Overexpression of AQPs 1, 3, 5 and 9 protein has been observed in ovarian tumors [52], and has been the major focus of papers published in the ovarian cancer field (Table 6). AQP1 protein expression was upregulated in the late stages of ovarian tumors, but the positive or negative associations with survival rates of patients depended on the cancer subtype category [52,187,188]. AQP3 upregulation by EGF promoted cell migration in an ovarian cancer cell line, which was inhibited by curcumin [189]. Inhibition of AQP3 by Auphen inhibited proliferation of xenografted hepatocarcinoma cells in mice [59,190]. Conversely, in patients with urothelial or bladder carcinomas, high levels of AQP3 protein and RNA were associated with better progression-free survival [60,191], suggesting that the roles of AQP classes are likely to depend on the cancer subtype. Multiple studies have reported direct correlations of AQP5 expression levels with tumor grade, lymph node metastasis and poor prognoses, suggesting AQP5 could be a prognostic factor for ovarian cancer [52,192,193]. AQP5 expression level was also associated with the sensitivity of ovarian cancer cells to chemotherapy [194]. AQP6 and AQP9 were reported to be downregulated, whereas AQP8 was unchanged, in ovarian cancer, but their roles remain to be determined [52,195].

Largely consistent with the systematic review results (Figure 2) which indicated high hazard ratios for AQP1 (HR 1.4), AQP5 (HR 1.6) and AQP9 (HR 1.9), though interestingly not AQP3, transcript levels obtained from the Human Protein Atlas database suggested high levels of AQPs 1, 3, 5 and 9 in ovarian cancer biopsies (Figure 3E), as well as a possible increase in AQP11. Increased transcript levels correlated with reduced overall survival times for ovarian cancer patients across the panel of AQPs (HRs 1.2 to 1.9), with the exception of AQPs 4, 7 and 8 (Figure 4E). Conversely, increased levels of AQP7 and AQP8 expression correlated with longer survival times, suggesting a possible protective role in cancer that could be analogous to effects of AQPs 4 and 11 in breast cancer, as noted above.

### 2.6. AQPs in Endometrial Cancer

The roles of AQPs in endometrial cancer and other less-well-studied cancers remain a gap in knowledge in the field (Table 7). Endometrial cancer arises from tissue lining the uterus (the endometrium) and accounts for about 95% of uterine cancers. Protein and RNA for AQPs 1, 2, 3, 5, 7 and 9 are normally expressed in endometrium, in which the AQPs are thought to be involved in fluid exchange and estrogen-mediated regulatory effects [200,201,202,203,204,205,206]. AQP1 expression in endometrial cancer has been correlated with histologic grade, extent of myometrial invasion and the likelihood of extrauterine metastasis [207], but the functional role of AQP1 in this tissue remains to be defined.

Data from the Human Protein Atlas database indicated that AQPs 1, 3 and 5 showed relatively higher levels of transcripts in endometrial cancer biopsies, as compared to other classes (Figure 3F). Increased transcript levels for AQP5 were correlated with poor survival, with hazard ratios of 1.4 in endometrial cancers (Figure 4F). Increased hazard ratios seen for AQP4 (HR1.5) and perhaps AQP9 (HR1.3) remain a gap in knowledge in the field. Future research could explore possible roles of classes of AQPs in endometrial and other cancers.

## 3. Discussion

Patterns of upregulation of specific classes of aquaporins were repeated for clusters of cancers, and they showed associations with the risk of death. The most frequent pattern linked to pathological severity involved AQPs 1, 3 and/or 5 as negative indicators for multiple cancer types. In glioblastoma, lung and ovarian cancers, AQP9 also appeared to be a candidate of interest for cancer severity. Conversely, a fascinating observation was that higher levels of AQPs 7 and 8 were associated with lower hazard ratios in glioblastoma and ovarian cancer; and AQP11 appeared to have a beneficial influence in breast and colorectal cancers. Possible protective mechanisms remain unexplored, but could involve possible roles, for example, for AQPs 8 or 9, in promoting the cell volume loss preceding apoptosis [35], or for AQPs 7 or 11, in enabling glycerol or hydrogen peroxide transport [285]. Although fundamental mechanisms of aquaporins both for promoting cancer metastasis and in exerting protective influences remain to be defined, results here support the hypothesis that overexpression of selected classes of AQPs differentially augments cancer progression, depending on the cancer subtype. Patterns of aquaporin expression also have been linked with differences in sensitivity to chemotherapy treatments, suggesting possible roles as biomarkers for designing targeted treatments. AQP5, in particular, underpinned a recurring theme as a biomarker in diverse cancers, including colon, breast and ovarian. AQP channels localized in the leading edges of migrating cancer cells are positioned to enhance cell migration as part of the volume regulatory engine driving process extension. Aquaporins merit investigation as cancer-specific therapeutic targets. Pending progress in defining subtype-selective pharmacological modulators of AQPs, tailored combinations of agents could be used to target specific cancer types and greatly expand clinical options for cancer treatment.

## 4. Materials and Methods

### 4.1. Systematic Review Protocol and Inclusion Criteria

Data included in the systematic analysis included all primary research published between 1 January 1990 and 1 January 2019 that was indexed in PubMed or Embase (OVID version) online databases and reported to have investigated aquaporins and cancers. Papers included work using biopsies, cancer cell lines, animal models and human patient cohort studies. Lists of the search keywords used for querying the online databases are summarized in Table 1.

For positive hits identified using the search keywords, titles and abstracts were screened to determine whether the retrieved studies met intended criteria. Review articles, conference abstracts and studies that did not have a focus on aquaporins were excluded. For studies that met the selection criteria, complete published papers were obtained and evaluated (flowchart in Figure 1; extracted data summary in Appendix A).

### 4.2. Forest Plot Analyses

Hazard ratios for overall survival rates in people with cancers were extracted from data in the final papers collected in the comprehensive literature search. Forest plots utilizing the random effects model were generated to assess hazard ratios for each class of AQPs, using Stata software (Stata software, StataCorp, College Station, TX, USA).

Overall survival times of people with cancers were obtained from the Human Protein Atlas database, available at https://www.proteinatlas.org/humanproteome/pathology [286,287]. Data from patient biopsy samples were classified into high or low expression groups (above or below median) for AQP classes 1 to 10, based on RNAseq data quantified as “fragments per kilobase of exon per million reads mapped” (FPKM) values for AQP transcript levels. RNAseq data were used to calculate hazard ratios for each class of AQPs classified by the type of cancer, using GraphPad Prism 8 (San Diego, CA, USA). Forest plots based on the random effects model were generated to determine hazard ratios for each class of AQPs, using Stata software.

### 4.3. Statistical Analyses

Box plots were generated by using GraphPad Prism 8 software to summarize transcript levels for classes of AQPs measured in samples of human glioma, colon cancer, lung cancer, breast cancer, ovarian cancer and endometrial cancer biopsies, compiled from Human Protein Atlas transcriptomic data. The median values for the transcript levels for all AQP classes in individual patient samples were used as the point of reference for statistical comparisons. Statistically significant outcomes determined by the non-parametric Mann–Whitney U tests (with GraphPad Prism 8) are reported as * *p* < 0.05. NS is not significant.

## 5. Conclusions

Overexpression of specific classes of AQPs is consistently observed in clinical and preclinical studies of cancers. Different classes of AQPs have been linked to properties of migration, invasion, proliferation and angiogenesis, depending on the cancer type. Analyses here provide evidence that the upregulation of certain AQPs is negatively associated with survival time for people with cancers. AQPs 1, 3, 5 and 9 in particular are associated with reduced survival for cases of glioma, ovarian and endometrial cancers, via both direct and indirect mechanisms yet to be defined. High hazard ratios were noted for AQP5 and AQP9 in glioma and ovarian cancers; more research is needed on their possible pathophysiological roles. Not all classes of AQPs were associated with worse outcomes; on the contrary, AQPs 7, 8 and 11 are intriguing as potential components of protective mechanisms against some cancer types, framing a novel gap in knowledge in the field. In summary, results here provide a logical rationale for evaluating AQPs as targets for tailored cancer therapies. Small molecule inhibitors of AQPs are being developed, though they remain to be advanced to clinical trials. Research to define and optimize AQP pharmacological agents, and studies to explore the mechanisms of these channels in cancer growth and progression, are needed to address a clinically important but untapped area of work.

## Figures and Tables

**Figure 1 cancers-12-01911-f001:**
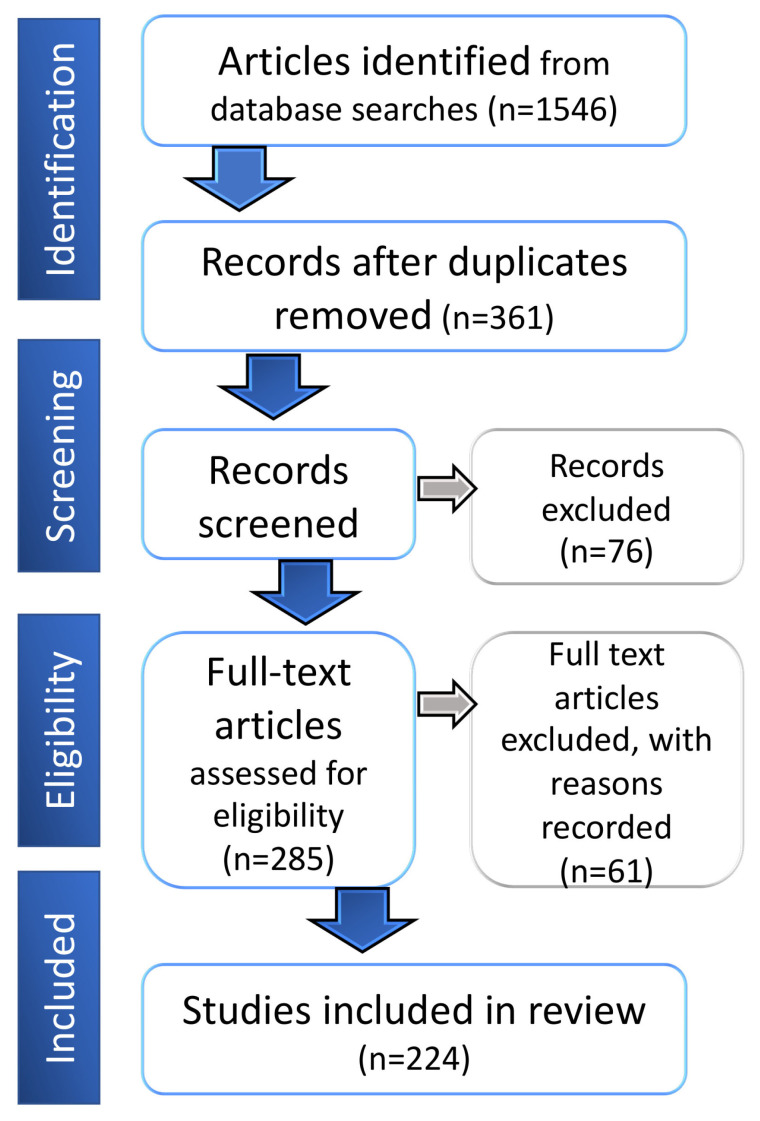
Flow diagram showing the process of the literature evaluation for the systematic review.

**Figure 2 cancers-12-01911-f002:**
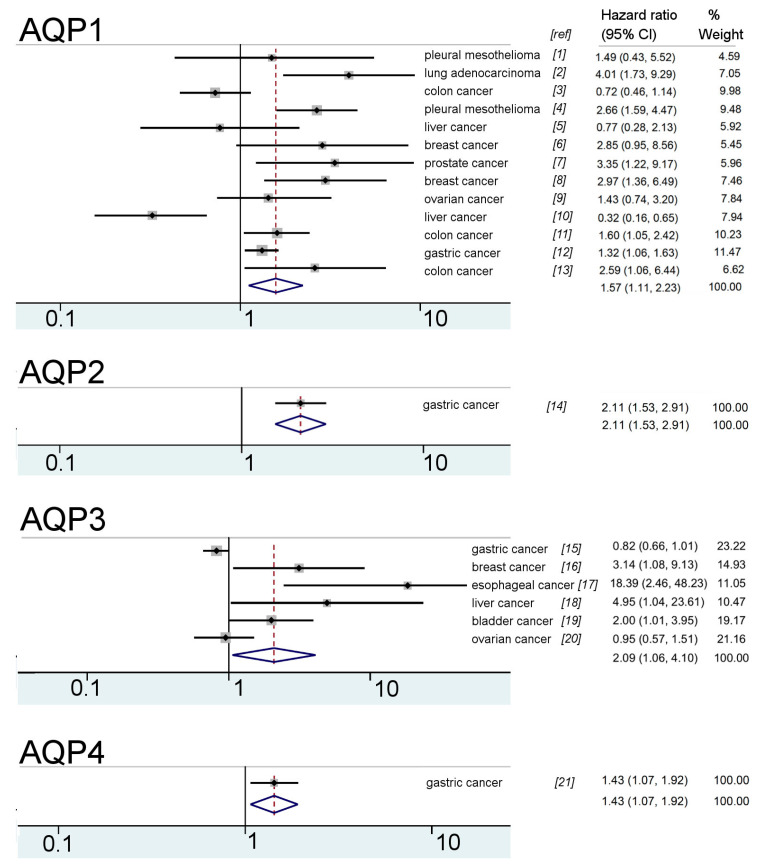
Forest plots compiled for aquaporin (AQP) classes, with hazard ratio and confidence interval (CI) data extracted from the published literature, based on analyses of levels of protein, RNA or both. References listed in the figure as [ref] for data sources are as follows: *[1]* Angelico, 2018 [45]; *[2,24]* Bellezza, 2017 [46]; *[3]* Kang, 2015 [47]; *[4]* Kao, 2012 [48]; *[5]* Luo, 2017 [44]; *[6]* Otterbach, 2010 [49]; *[7]* Park, 2017 [50]; *[8]* Qin, 2016 [51]; *[9,20,23,29]* Sato, 2018 [52]; *[10]* Sekine, 2014 [53]; *[11]* Smith, 2019 [54]; *[12,14,15,21,22,25,26,27,30]* Thapa, 2018 [55]; *[13]* Yoshida, 2013 [56]; *[16]* Chae, 2015 [57]; *[17]* Liu, 2013 [58]; *[18,28]* Peng, 2016 [59]; *[19]* Rubenwolf, 2015 [60]. Italicized numbers under the heading *“[ref]”* in this Figure correspond to the citations designated in this figure legend.

**Figure 3 cancers-12-01911-f003:**
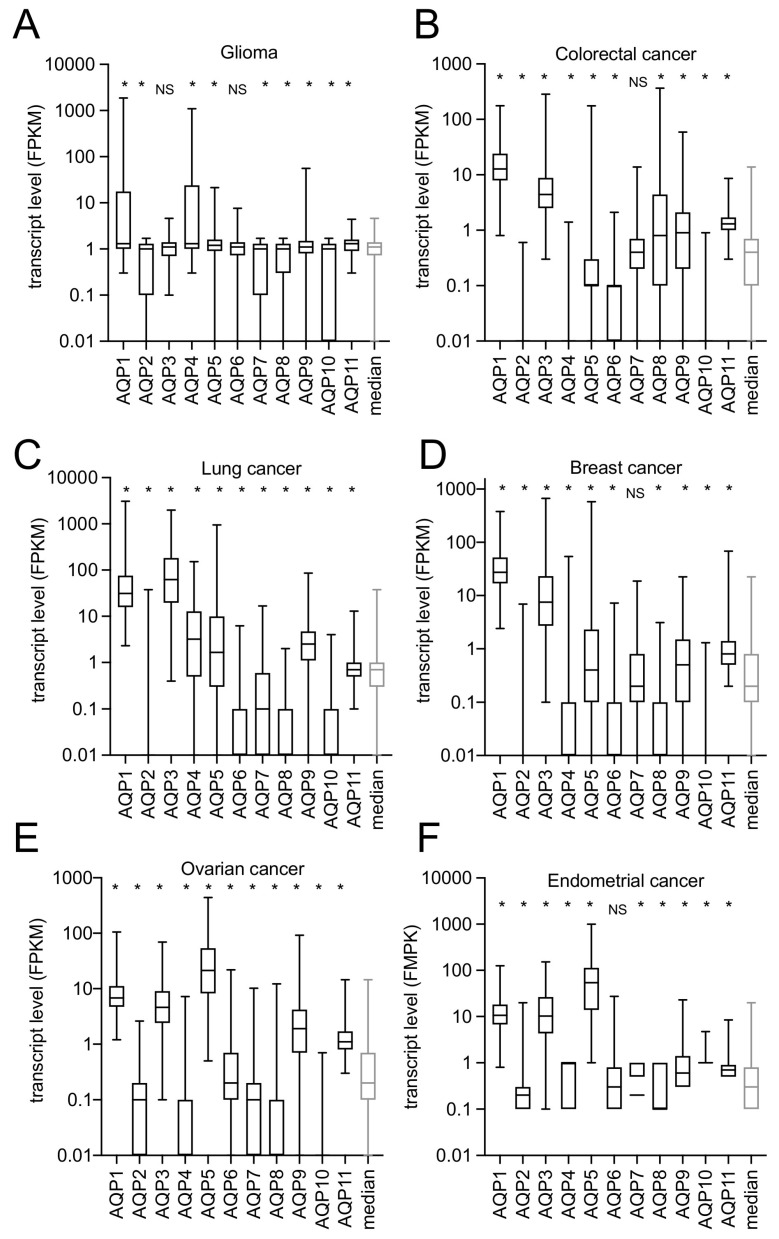
Quantitative transcript levels in human cancer biopsies, calculated as “fragments per kilobase of exon per million reads mapped” (FPKM). Data from the RNAseq transcriptomic database (Human Protein Atlas, https://www.proteinatlas.org), summarized by AQP class, are shown as box plots for six cancer types (**A**–**F**, as indicated). Boxes show 50% of data points; error bars show the full range; horizontal bars show median values. Median transcript levels are average FPMK values for all classes of AQPs (1–11) within each cancer type. * *p* < 0.05 as compared with the median values (Mann Whitney U-test); NS is not significant.

**Figure 4 cancers-12-01911-f004:**
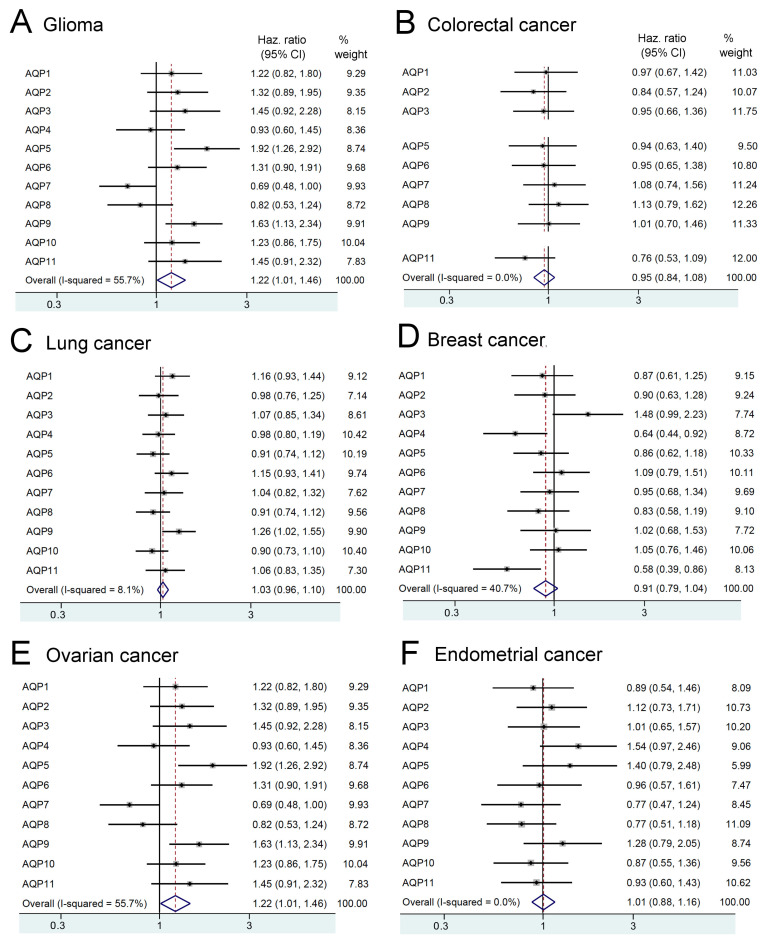
Increases and decreases in the risk of death for cancer patients show patterns associated with the upregulation of transcript levels for different classes of AQPs, depending on cancer type. Forest plots depict associations between survival time and median RNA expression compiled for multiple samples (patient data from the Human Protein Atlas) for six types of cancers (**A**–**F**). Hazard ratios estimate the magnitude of the effect, calculated using an interactive algorithm on the Atlas site that divides populations into high and low expression groups by median FPKM values. The vertical line indicates no effect (odds ratio 1.0); horizontal lines indicate 95% confidence interval (CI); the percent weight reflects the power of the analysis (increased by higher n values and tighter confidence intervals).

**Table 1 cancers-12-01911-t001:** Search terms used for data collection. (Searches combined the left and right sets with AND).

Embase	
Neoplasm/exp OR cancer *: tiab OR neoplasm *: tiab OR Metastas *: tiab OR Tumor *: tiab OR Tumor *: tiab	Aquaporin/de OR “Aquaporin1”/de OR “Aquaporin2”/de OR “Aquaporin3”/de OR “Aquaporin4”/de OR “Aquaporin5”/de OR “Aquaporin6”/de OR “Aquaporin7”/de OR “Aquaporin8”/de OR “Aquaporin9”/de OR “Aquaporin10”/de OR “Aquaporin11”/de OR “Aquaporin12”/de OR “Aquaporin0”/de OR Aquaporin *: tiab OR “Water channel *”: tiab OR AQP *: tiab OR CHIP28 *: tiab
**PubMed**	
Neoplasms[mh] OR cancer *[tiab] OR neoplasm *[tiab] OR “Neoplasm Metastasis” [mh] OR Metastas *[tiab] OR Tumor *[tiab] OR Tumor *[tiab]	Aquaporins[mh] OR Aquaporin *[tiab] OR “Water channel *”[tiab] OR AQP *[tiab] OR CHIP28 *[tiab] OR MIP *[tiab]

* The asterisk is a wildcard symbol used to broaden search terms for literature database queries.

**Table 2 cancers-12-01911-t002:** Publications on AQPs in glioma, classified by research approach.

AQP Class	# Pubs	Research Approach (with Cited Papers in Each Category)	
		Expression	Function
AQP1	21	[49,66,67,68,69,70,71,72,73,74,75,76,77,78,79,80,81,82,83,84]	[73,74,75,77,81,83]
AQP2	1	[79]	NIF
AQP3	2	[79,85]	[85]
AQP4	25	[78,79,81,82,84,86,87,88,89,90,91,92,93,94,95,96,97,98,99,100,101,102,103,104,105]	[81,87,88,89,90,102,103]
AQP5	2	[79,106]	[106]
AQP6	1	[79]	NIF
AQP7	1	[79]	NIF
AQP8	2	[79,107]	NIF
AQP9	5	[79,104,108,109,110]	[110]
AQP10	1	[79]	NIF
AQP11	1	[79]	NIF

NIF: none identified in final set of papers evaluated. “# Pubs” refers to the number of publications that met criteria for inclusion in the systematic review of literature; see Results text for more details.

**Table 3 cancers-12-01911-t003:** Publications on AQPs in colon cancers, classified by research approach.

AQP Class	# Pubs	Research Approach (with Cited Papers in Each Category)	
		Expression	Function
AQP1	11	[56,123]	[119,120,121,122,123,130]
AQP2	0	NIF	NIF
AQP3	2	[129,131]	[129]
AQP4	0	NIF	NIF
AQP5	11	[124,125,126,127,128,129,131,132,133,134,135]	[126,127,128,129,132,133,134,135]
AQP6	0	NIF	NIF
AQP7	0	NIF	NIF
AQP8	1	[136]	[136]
AQP9	3	[137,138,139]	[138], [139]
AQP10	0	NIF	NIF
AQP11	0	NIF	NIF

NIF: none identified in final set of papers evaluated. “# Pubs” refers to the number of publications that met criteria for inclusion in the systematic review of literature; see Results text for more details.

**Table 4 cancers-12-01911-t004:** Publications on AQPs in lung cancers, classified by research approach.

AQP Class	# Pubs	Research Approach (with Cited Papers in Each Category)	
		Expression	Function
AQP1	12	[46,99,117,142,143,145,146,147,155,156,157,158]	[99,145,156,157,158]
AQP2	0		NIF
AQP3	8	[142,148,155,159,160,161,162,163]	[159,161,162]
AQP4	2	[101,155]	NIF
AQP5	13	[46,142,147,149,150,151,152,153,154,155,160,164,165]	[149,152,153,154,164,165]
AQP6	0	NIF	NIF
AQP7	0	NIF	NIF
AQP8	0	NIF	NIF
AQP9	1	[166]	[166]
AQP10	0	NIF	NIF
AQP11	1	[167]	NIF

NIF: none identified in final set of papers evaluated. “# Pubs” refers to the number of publications that met criteria for inclusion in the systematic review of literature; see Results text for more details.

**Table 5 cancers-12-01911-t005:** Publications on AQPs in breast cancer, classified by research approach.

AQP Class	# Pubs	Research Approach (with Cited Papers in Each Category)	
		Expression	Function
AQP1	6	[51,117,171,180,181]	[51,175,181]
AQP2	1	[180]	NIF
AQP3	8	[47,57,85,173,174,175,180]	[85,159,173,174,175]
AQP4	2	[180,182]	NIF
AQP5	7	[176,177,178,179,180,183,184]	[178,179,183]
AQP6	1	[180]	NIF
AQP7	1	[180]	NIF
AQP8	1	[180]	NIF
AQP9	2	[85,180]	[85]
AQP10	1	[180]	NIF
AQP11	1	[180]	NIF

NIF: none identified in final set of papers evaluated. “# Pubs” refers to the number of publications that met criteria for inclusion in the systematic review of literature; see Results text for more details.

**Table 6 cancers-12-01911-t006:** Publications on AQPs in ovarian cancer, classified by research approach.

AQP Class	# Pubs	Research Approach (with Cited Papers in Each Category)	
		Expression	Function
AQP1	6	[52,117,188,194,196,197]	[188]
AQP2	3	[188,197]	[188]
AQP3	8	[52,85,187,188,189,197,198]	[85,188,189]
AQP4	3	[188,197]	[188]
AQP5	8	[52,188,192,193,194,197,199]	[188,194,199]
AQP6	4	[120,188,197]	[188]
AQP7	4	[187,188,197]	[188]
AQP8	4	[188,195,197]	[188]
AQP9	6	[52,85,187,188,197]	[85,188]
AQP10	2	[188]	[188]
AQP11	0	NIF	NIF

NIF: none identified in final set of papers evaluated. “# Pubs” refers to the number of publications that met criteria for inclusion in the systematic review of literature; see Results text for more details.

**Table 7 cancers-12-01911-t007:** Publications on AQPs in other cancer types.

Cancer Type	Type of AQP	Cited References
Biliary tract cancer	AQP1	[53]
	AQP5	[172]
Bladder cancer	AQP1	[208]
	AQP3	[60]
Bone cancer	AQP3	[209]
Cervical cancer	AQP1	[154,210,211,212,213,214]
	AQP3	[211,213,214]
	AQP4	[213]
	AQP5	[213,215,216]
	AQP8	[210,213,214,217,218,219]
Endometrial cancer	AQP1	[207]
	AQP2	[220]
Gallbladder cancer	AQP5	[221]
Gastric cancer	AQP1-11	[55]
	AQP2	[222]
	AQP3	[223,224,225,226,227]
	AQP5	[228]
Leukemia	AQP5	[229]
	AQP8	[230]
	AQP9	[231,232,233,234]
Liver cancer	AQP1	[144,235]
	AQP3	[59,196,236,237,238]
	AQP5	[238,239,240,241]
	AQP7	[237]
	AQP9	[59,196,237,242,243,244]
Melanoma, cutaneous	AQP1	[245,246]
Mesothelioma	AQP1	[45,48,247,248,249,250,251,252]
Esophageal cancer	AQP3	[58]
	AQP4	[253]
	AQP5	[58,254]
	AQP8	[255]
Pancreatic cancer	AQP1	[256]
	AQP3	[256,257]
Prostate cancer	AQP1	[50,117,258,259]
	AQP2	[260]
	AQP3	[258,261,262]
	AQP5	[263]
Renal cancer	AQP1	[264,265,266,267,268,269,270,271]
	AQP5	[272]
Skin cancer	AQP1	[273,274,275,276]
	AQP3	[161,277,278,279]
Squamous cell carcinoma, oral	AQP3	[280]
Squamous cell carcinoma, pharyngeal	AQP1, 5	[281]
Squamous cell carcinoma, tongue	AQP3, 5	[162]
Thyroid cancer	AQP3, 4	[282]
Urothelial Carcinoma	AQP1	[283]
	AQP3	[197,284]

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
