# Peer review of "Combined Systematic Review and Transcriptomic Analyses of Mammalian Aquaporin Classes 1 to 10 as Biomarkers and Prognostic Indicators in Diverse Cancers"

_cancers, 2020, doi:10.3390/cancers12071911_

Round 1

Reviewer 1 Report

Review of AJ Yool Aquaporin Review in Cacners

The majority of this review article is an impressive encyclopedic database of the known relationships between aquaporin classes (AQP0-AQP12) and cancers.  This will be a valuable reference to aquaporin researchers everywhere. 

However, there is little here that will capture the interest of cancer biologists.  What is missing is some discussion, either in the introduction or the conclusion section, of the known (and speculated) differences in the activities of the different members of the AQP family.   There was the beginning of a discussion about AQP and invadopodia, but then this was truncated.  And expansion fo this would be warranted.  Without that being addressed, the differences in expression cannot be interpreted with any biological framework.  This would be a great opportunity for speculation. 

Reviewer 2 Report

The manuscript by Pak Hin Chow and collegues is a interesting combined systemic review and transcriptomic analysis of RNAseq data, focusing on AQP in cancer.

The manuscript is accepted in the present form.

Reviewer 3 Report

In the article "Combined systematic review and transcriptomic analyses of mammalian aquaporin classes 1 to 10 as biomarkers and prognostic indicators in diverse cancers." authors study published scientific findings on the effects of aquaporin genes on cancers, and analyze the statistical associations of those genes at the RNA level with patient overall survival.

Although sample RNA concentration data from TCGA are accessible from many sources, authors choose the protein atlas. "Protein" is referred to throughout the text, without to provide any separate analysis of protein associations on prognosis. A greater emphasis must be made on the type of macromolecule that is analyzed and presented in each sentence/figure.

In the main text, for each of the primary cancers, effects of aquaporin gene products are presented. This is fine, as long as the criteria for grouping them are presented rigorously, clearly and consistently.

The best section of the manuscript is the conclusions paragraph (NOT the conclusions that are presented in the main text, which still need to become better integrated). It would be desirable to edit the rest of the manuscript with aim to reach the same clarity.

Optional: If possible, the abstract should be shortened and focused on the main findings of the study.

Potential suggestions:

A. Restructure the article. Either one of the three following approaches can be used.

- The first approach is to add after the introduction an entire new section that explains the rationale for treating several members of the gene family together, by describing clearly the overarching principles which link them to carcinogenesis.

- The second approach is to divide Aquaporins in groups according to their effect on cancer progression.

For each Aquaporin, a section or paragraph must be created, with a clear separation between RNA and protein information, and a section with a discussion of their effects. After this clear separation, possible unifying principles can be discussed.

- The third approach is to divide some of the sections into more paragraphs, according to their take-home message.

B. In the discussion, one option is to compile separately the good and the poor prognosis -associated Aquaporins. There, a possible explanation can be attempted, based on their biological properties.

Minor: Correct the legend for figure 4. As the materials section explains, FPKM corresponds to RNA and not Protein, which is nowhere in the legend mentioned. A figure legend has to be independent from the materials section, and leave no room for misunderstanding.

Supplement, ref 191. correct "xeongrafted". Further typing errors can be corrected after proofreading.

Round 2

Reviewer 3 Report

The article has been improved.